# Prognostic factors for severity and mortality in patients infected with COVID-19: A systematic review

Ariel Izcovich[1]*, Martín Alberto Ragusa[2], Fernando Tortosa[3], María Andrea Lavena Marzio[1], Camila Agnoletti[1], Agustín Bengolea[1], Agustina Ceirano[1], Federico Espinosa[1], Ezequiel Saavedra[1], Verónica Sanguine[4], Alfredo Tassara[1], Candelaria Cid[1], Hugo Norberto Catalano[1], Arnav Agarwal[5], Farid Foroutan[6], Gabriel Rada[7,8,9]

1 Servicio de clínica médica, Hospital Alemán, Buenos Aires, Argentina, 2 Servicio de clínica médica, Hospital Fernández, Buenos Aires, Argentina, 3 Departamento Médico, Hospital "Ramón Carrillo", San Carlos de Bariloche, Argentina, 4 Dirección Nacional de Calidad en Servicios de Salud y Regulación Sanitaria, Ministerio de Salud de la Nación, Buenos Aires, Argentina, 5 Department of Medicine, University of Toronto, Toronto, Ontario, Canada, 6 Ted Rogers Centre for Heart Research, University Health Network, Toronto, Canada, 7 Fundación Epistemonikos, Santiago, Chile, 8 UC Evidence Center, Cochrane Chile Associated Center, Pontificia Universidad Católica de Chile, Santiago, Chile, 9 Internal Medicine Department, Faculty of Medicine, Pontificia Universidad Católica de Chile, Santiago, Chile

* ariel.izcovich@gmail.com

**Data Availability Statement:** All relevant data are within the manuscript and its Supporting Information files.

## Abstract

### Background and purpose

The objective of our systematic review is to identify prognostic factors that may be used in decision-making related to the care of patients infected with COVID-19.

### Data sources

We conducted highly sensitive searches in PubMed/MEDLINE, the Cochrane Central Register of Controlled Trials (CENTRAL) and Embase. The searches covered the period from the inception date of each database until April 28, 2020. No study design, publication status or language restriction were applied.

### Study selection and data extraction

We included studies that assessed patients with confirmed or suspected SARS-CoV-2 infectious disease and examined one or more prognostic factors for mortality or disease severity.

Reviewers working in pairs independently screened studies for eligibility, extracted data and assessed the risk of bias. We performed meta-analyses and used GRADE to assess the certainty of the evidence for each prognostic factor and outcome.

### Results

We included 207 studies and found high or moderate certainty that the following 49 variables provide valuable prognostic information on mortality and/or severe disease in patients with

**Funding:** The author(s) received no specific funding for this work

**Competing interests:** The authors have declared that no competing interests exist.

COVID-19 infectious disease: Demographic factors (age, male sex, smoking), patient history factors (comorbidities, cerebrovascular disease, chronic obstructive pulmonary disease, chronic kidney disease, cardiovascular disease, cardiac arrhythmia, arterial hypertension, diabetes, dementia, cancer and dyslipidemia), physical examination factors (respiratory failure, low blood pressure, hypoxemia, tachycardia, dyspnea, anorexia, tachypnea, haemoptysis, abdominal pain, fatigue, fever and myalgia or arthralgia), laboratory factors (high blood procalcitonin, myocardial injury markers, high blood White Blood Cell count (WBC), high blood lactate, low blood platelet count, plasma creatinine increase, high blood D-dimer, high blood lactate dehydrogenase (LDH), high blood C-reactive protein (CRP), decrease in lymphocyte count, high blood aspartate aminotransferase (AST), decrease in blood albumin, high blood interleukin-6 (IL-6), high blood neutrophil count, high blood B-type natriuretic peptide (BNP), high blood urea nitrogen (BUN), high blood creatine kinase (CK), high blood bilirubin and high erythrocyte sedimentation rate (ESR)), radiological factors (consolidative infiltrate and pleural effusion) and high SOFA score (sequential organ failure assessment score).

## Conclusion

Identified prognostic factors can help clinicians and policy makers in tailoring management strategies for patients with COVID-19 infectious disease while researchers can utilise our findings to develop multivariable prognostic models that could eventually facilitate decision-making and improve patient important outcomes.

## Systematic review registration

Prospero registration number: CRD42020178802. Protocol available at: https://www.medrxiv.org/content/10.1101/2020.04.08.20056598v1.

## Introduction

COVID-19 is an infectious disease caused by the SARS-CoV-2 coronavirus [1]. It was first identified in Wuhan, China, on December 31, 2019 [2]; five months later, more than six million cases had been identified across 215 countries [3]. On March 11, 2020, WHO characterised the COVID-19 outbreak as a pandemic [1]. While the majority of cases present with mild symptoms, a minority progress to acute respiratory illness and hypoxia requiring hospitalization, and a subset develop acute respiratory distress syndrome, multi-organ failure or have fatal outcomes [4]. The case fatality rate reported across countries, settings and age groups is highly variable, ranging from about 0.5% to 10% [5]. In hospitalised patients it has been reported to be higher than 20% [6].

Prognostic factors (stand-alone or combined in risk assessment models) may guide the stratification of patients with SARS-CoV-2 infectious disease based on their risk of severe disease or death. This risk stratification may subsequently guide optimised management and resource utilisation strategies in the care of these patients.

Although multiple prognostic factors have been proposed and some have been accepted as "established" by the scientific community (i.e age), the predictive value of most of these potential prognostic factors has not been robustly evaluated and remains uncertain. As pointed out

by Wynants et.al: "unreliable predictors could cause more harm than benefit in guiding clinical decisions" [7]. For example, aggressive and risky interventions might be attempted if the risk of poor outcomes are inaccurately defined as high based on unreliable predictors.

Using innovative and agile processes supported by advanced evidence synthesis tools and collaborative efforts across several international research groups, this systematic review aims to provide a rigorous summary of the evidence available on prognostic factors that may be used in decision-making related to the care of patients infected with COVID-19.

## Methods

### Protocol registration

We published [8] and registered the protocol for this systematic review with PROSPERO (CRD42020178802).

### Search strategy

We conducted highly sensitive searches in PubMed/MEDLINE, the Cochrane Central Register of Controlled Trials (CENTRAL) and Embase. The searches covered the period from the inception date of each database until April 28, 2020. No study design, publication status or language restriction were applied.

Detailed strategies for each database are reported in the S1 Text.

In order to identify articles that might have been missed in the electronic searches, we reviewed the reference list of each included study and performed cross citation in Google Scholar using each included study as the index reference.

### Study selection

Four reviewers working independently and in duplicate, performed study selection, including screening of titles and abstracts and of potentially eligible full-text articles. Reviewers resolved disagreements by discussion.

We included studies examining individual prognostic factors or risk assessment models based on the typologies of prognosis proposed by Iorio and colleagues [9] and the PROGnosis RESearch Strategy (PROGRESS) Group framework [10] without applying any restrictions based on analytical methods (i.e performing multivariable analysis).

Specifically, we included studies that evaluated patients with confirmed SARS-CoV-2 infectious disease regardless of the healthcare setting (i.e ambulatory or inpatients) and of baseline disease severity. We investigated all candidate prognostic factors reported by individual studies and compared patients exposed (the factor was present) with patients unexposed (the factor was absent) to each one of those factors. We considered studies that assessed mortality or severe COVID-19 disease as outcomes and accepted the author's definitions of the latter. When severe COVID-19 disease was reported as a multi-categorical scale, we used the most severe category. Additionally, when severe COVID-19 disease was not reported as an outcome, we considered ICU requirement, invasive mechanical ventilation (IVM) and acute respiratory distress syndrome (ARDS) as surrogate outcomes.

### Data extraction

For each eligible study, five pairs of reviewers, independently, abstracted the following information on study characteristics (year of publication, country, medical center and time period in which the study was conducted); population characteristics (sample size, context in which the study was conducted and other population characteristics); description of prognostic

factors and outcomes and their definitions; and study results (measures of association or crude event rates for every candidate prognostic factor and outcome reported).

## Risk of bias assessment

Two reviewers assessed the risk of bias of individual included studies independently and in duplicate. Discrepancies were resolved by consensus. We used the Quality in Prognosis Studies (QUIPS) tool for prognostic factor studies [11] which considers population characteristics, attrition, prognostic factor and outcome measurement and potential residual confounding. For "study confounding summary" and "statistical analysis and presentation domains", in order to assess adequacy of the multivariable models, we considered appropriate model adjustment as based on inclusion of age, one comorbidity (e.g diabetes) and one parameter of disease severity (e.g. respiratory rate) at minimum.

## Data synthesis and analysis

We presented the results of individual prognostic factors in both tabular and narrative formats. We standardized the units of measurement for each prognostic factor, unifying the direction of the predictors and adjusting the weights of the studies, and calculated crude effect estimates when not provided [12]. When possible, we meta-analysed all prognostic factors whose association with the selected outcomes of interest was explored and were reported by more than one study. We used the generic inverse variance-based method to produce an overall measure of association and random-effect models based on the DerSimonian-Laird method provided by the metafor package for R software [13].

For every candidate prognostic factor, we presented the measure of association as odds ratios (ORs) and their corresponding 95% confidence intervals (CI). In studies that reported the measure of association as a hazard ratio (HR) or risk ratio (RR), we converted them to ORs using the baseline risk (death rate or incidence of severe COVID-19 out of the total sample) reported in the studies [14,15]. For dichotomous variables, we used the crude event rate to calculate ORs when no measures of association were provided. We excluded information on continuous variables for which no measures of association were available. We also calculated absolute risk differences (RDs) that can be attributed to every individual candidate prognostic factor by applying the ORs to estimated baseline risks (see below "Baseline risks"). When the same candidate prognostic factor was assessed in alternative ways (e.g. dichotomic and continuous) we used the one for which we found better certainty of evidence.

For every explored candidate variable, we performed sensitivity analysis excluding high risk of bias studies and studies that did not report adjusted estimates. In cases where the effect estimates provided by the primary analysis and the sensitivity analysis significantly differed, we either presented the moderate/low risk of bias–adjusted estimates or the primary analysis estimates but rated down certainty of the evidence because for risk of bias (see below). In addition, when we observed inconsistent results for disease severity outcome, we performed subgroup analyses accounting for outcome definition (i.e severity scale vs. IVM vs. ARDS) as a potential source of heterogeneity.

## Assessment of certainty of the evidence

We assessed certainty of the evidence for each candidate prognostic factor, by outcome, based on the GRADE approach [16]. The approach considers the following domains: risk of bias, indirectness, inconsistency, imprecision, and publication bias. We produced summary of findings tables and rated the certainty of the evidence as high, moderate, low or very low

depending on the grading of the individual domains [16]. See S1 Text for a detailed description of the certainty of the evidence assessment.

## Result interpretation

To define which candidate variables provide valuable prognostic information we adopted a minimally contextualized approach [17]. To this end, we arbitrarily set thresholds to define important incremental increase in the risk of our outcomes. In setting those thresholds we aimed to define the minimal incremental increase in the risk of mortality or severe COVID-19 disease that could be interpreted as valuable prognostic information without considering the potential consequences of using that information in healthcare decision-making. These thresholds represent the line that separates a risk increase that is trivial from a small but important risk increase. We set those thresholds in 0.5% increase in mortality and 1% increase in severe COVID-19 disease. We performed a sensitivity analysis in which we adopted a purely non-contextualized approach [17] to assess mortality outcome. In doing so we only considered the relative measures of association and used an OR of 1 as the threshold for minimal important risk increment.

## Baseline risks

To define baseline risks we selected clinical scenarios based in the severity categories proposed by WHO [18]. To assess the prognostic value on mortality, we used the clinical scenario of a patient infected with COVID-19 with severe but not critical disease (i.e patients with respiratory failure but not invasive mechanical ventilation and/or hemodynamic support requirement). We identified one study informing prognosis in this specific subgroup with a mortality risk of 9% [19]. However, as we identified significant variability in mortality risks reported for similar clinical scenarios (i.e in the RECOVERY trial [20] mortality risk in hospitalized patients assigned to the control arm, with no baseline oxygen requirement was 14%), we performed a sensitivity analysis using a baseline risk of 26% as reported by a large cohort of non-ICU inpatients treated in 255 sites across 36 countries [6].

To assess the prognostic value on severe COVID-19 disease, we used the clinical scenario of a patient infected with COVID-19 with non-severe disease. We identified 7 studies informing prognosis in this specific subgroup with a median risk of progression to severe or critical state of 13% [21–27]. We calculated baseline risks (risks in patients not exposed to the prognostic factor) by also considering the prevalence of every prognostic factor and the estimates of association [28]. When prevalence of prognostic factors was not available we used described baseline risks (9% for mortality and 13% for severe COVID-19 disease).

## Update of this systematic review

An artificial intelligence algorithm deployed in the Coronavirus/COVID-19 topic of the L. OVE platform (https://app.iloveevidence.com/loves/5e6fdb9669c00e4ac072701d) will provide instant notification of articles with a high likelihood of eligibility. These will be screened by paired reviewers iteratively and will conduct data extraction and iterative updates of estimates for selected prognostic factors accordingly. We will consider resubmission to a journal if there is a substantial modification on the measure of association or the certainty of the evidence for a given prognostic factor such that it is clinically significant, at the discretion of the reviewer team. This review is part of a larger project established to produce multiple parallel systematic reviews relevant to COVID-19 [29].

## Results

(Fig 1) illustrates the Preferred Reporting Items for Systematic Reviews and Meta-Analyses (PRISMA) flow diagram. Our search identified 7631 citations of which we included 569 studies for full text assessment. 207 studies fulfilled the inclusion criteria [21,23–25,27,30–231]. These 207 studies, with sample sizes from 10 to 8910, enrolled a total of 75607 patients and were conducted in 12 different countries (China, USA, Canada, Spain, France, Turkey, Korea, Japan, Italy, Germany, India and Singapore).

### Description of included studies

S1 Table describes the characteristics of the included studies reporting on mortality and/or severe COVID-19 disease. Regarding candidate prognostic factors, of the 207 included studies, 184 (88.9%) reported socio-demographic variables, 180 (86.9%) comorbidities, 178 (86%) clinical findings, 176 (85%) laboratory findings and 106 (51.3%) imaging findings. The outcomes reported were mortality in 116 (56%) and progression to severe/critical status in 131 (63.3%). In 78 (37.7%) of the included studies a multivariable analysis was performed. In the 150 studies in which the severity of included patients was described the mean proportion of patients in each category was: non-severe disease 63.8%, severe disease 22.6%, critical disease 13.6%.

### Risk of bias

Risk of bias was high across most identified studies. Among the 207 included studies only 7 were judged as low risk of bias [21,24,56,57,59,130,142] as the remaining presented important

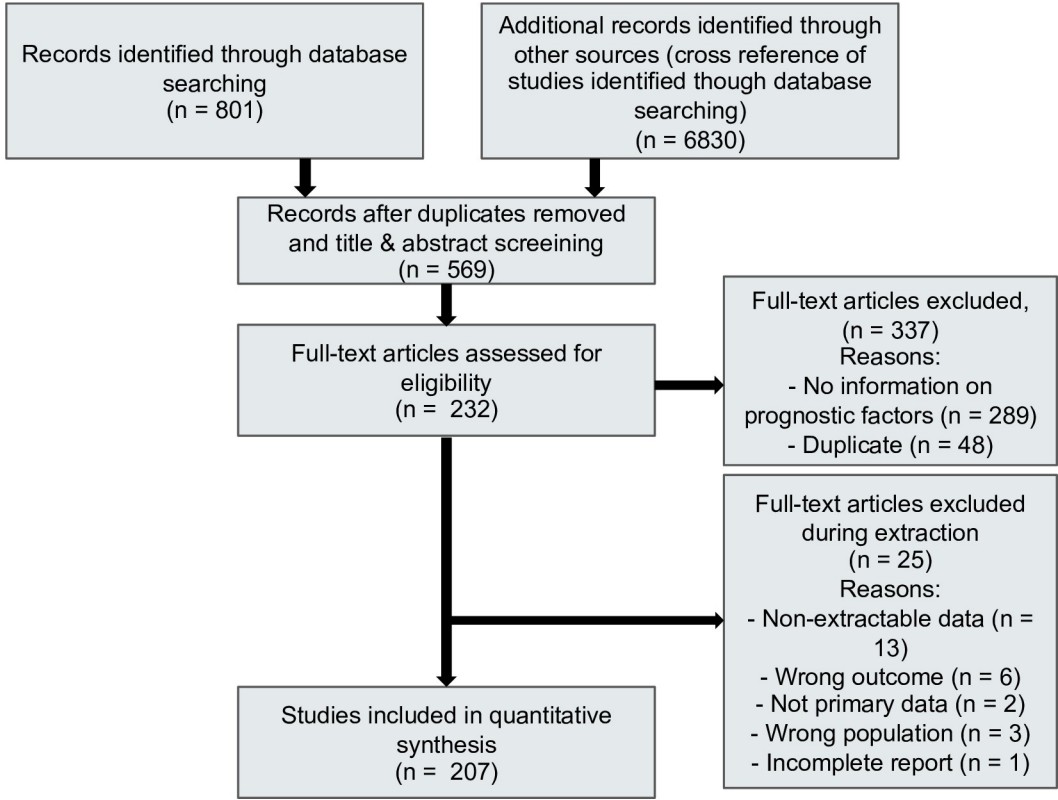

**Fig 1. PRISMA (preferred reporting items for systematic reviews and meta-analyses) flowchart of study inclusions and exclusions.**

limitations in at least one domain or item. Most frequent were retrospective design, which may have introduced classification bias, and lack or inappropriate adjusted analysis. S2 Table provides detailed judgements for each of the risk of bias domains criteria.

## Prognostic factors for mortality

We investigated 96 candidate prognostic factors for mortality from 116 studies including 57044 patients. S3 Table provides a summary of findings for all the candidate prognostic factors and S1 Appendix includes the corresponding forest plots.

We found high or moderate certainty that the following 35 variables provide valuable prognostic information on mortality outcome (Table 1).

**Demographic factors.** Age per 10 years increase (OR 1.8, 95% CI 1.54 to 2.1; RD 6.1%, 95% CI 4.2 to 8.2%), male sex (OR 1.72, 95% CI 1.5 to 1.98; RD 5%, 95% CI 4 to 7%) and active smoker (OR 1.57, 95% CI 1.19 to 2.07; RD 4.3%, 95%CI 1.5 to 7.5%).

**Medical illness and patient history factors.** Any chronic condition or comorbidity (OR 3.3, 95% CI 2.18 to 5; RD 10.3%, 95% CI 6.8 to 13.4%), cerebrovascular disease (OR 2.85, 95% CI 2.02 to 4.01; RD 12.6%, 95% CI 7.5 to 18.5%), chronic obstructive pulmonary disease (COPD) (OR 2.43, 95% CI 1.88 to 3.14; RD 9.8%, 95% CI 6.4 to 13.6%), chronic kidney disease (CKD), (OR 2.27, 95% CI 1.69 to 3.05; RD 8.8%, 95% CI 5.1 to 12.9%), cardiovascular disease (defined as coronary heart disease and/or cardiac failure) (OR 2.12, 95% CI 1.77 to 2.56; RD 7.5%, 95% CI 5.4 to 9.7%), cardiac arrhythmia (OR 2.13, 95% CI 1.72 to 2.65; RD 6.5%, 95% CI 4.7 to 8.4%), arterial hypertension (OR, 2.02, 95% CI 1.71 to 2.38; RD 6%, 95% CI 4.5 to 7.33), diabetes (OR 1.84, 95% CI 1.61 to 2.1; RD 5.6%, 95% CI 4.3 to 7%), dementia (OR 1.54, 95% CI 1.31 to 1.81; RD 4.2%, 95% CI 2.5 to 6.2%), obesity (OR 1.41, 95% CI 1.15 to 1.74; RD 3.1%, 95% CI 1.2 to 5.1%), cancer (OR 1.35, 95% CI 1.17 to 1.55; RD 2.7%, 95% CI 1.4 to 4.2) and dyslipidemia (OR 1.26, 95%CI 1.06–1.5; RD 2.1%, 95%CI 0.5% to 3.9%).

**Symptoms, vital signs and physical examination factors.** Respiratory failure (OR 21.17, 95% CI 4.9 to 91.3; RD 20.3%, 95% CI 13.4% to 22.4%), low blood pressure (OR 6.7, 95% CI 3.14 to 14.33; RD 30.9%, 95% CI 14.7% to 49.6%), hypoxemia (OR 5.46, 95% CI 2.05 to 14.53; RD 6.7%, 95% CI 4.2% to 7.7%), tachycardia (OR 2.61, 95% CI 1.62 to 4.22; RD 11.5%, 95% CI 4.8% to 20.4%), dyspnea (OR 3.45, 95% CI 2.72 to 4.38; RD 8.9%, 95% CI 7.5% to 10.2%), anorexia (OR 2.16, 95% CI 1.14 to 4.12; RD 7%, 95% CI 1.1% to 13.1%) and tachypnea (OR 1.21, 95%CI 1.12 to 1.31; RD 1.4%, 95% CI 0.9% to 1.9%).

**Laboratory factors (measured in blood or plasma).** High procalcitonin (OR 12.42, 95% CI 7.18 to 21.5; RD 32.3%, 95% CI 25% to 38.1%), myocardial injury markers (OR 10.89, 95% CI 5.39 to 22.04; RD 16.9%, 95% CI 13.4% to 19%), high white cell count (WBC) (OR 4.06, 95% CI 2.7 to 6.12; RD 16.9%, 95% CI 11% to 23.3%), high lactate (OR 3.66, 95% CI 2.26 to 5.94; RD 14.3%, 95% CI 8.3% to 20.6%), low platelet count (OR 5.43, 95% CI 2.55 to 11.56; RD 14.3%, 95% CI 8.3% to 18.6%), high D-dimer (OR 4.81, 95% CI 3.15 to 7.34; RD 11.2%, 95% CI 8.8% to 13.1%), high lactate dehydrogenase (LDH) (OR 4.09, 95% CI 1.18 to 14.17; RD 10.4%, 95% CI 1.4% to 15.3%), high c-reactive protein (CRP) (OR 6.6, 95% CI 3.36 to 12.99; RD 7.9%, 95% CI 6.4% to 8.7%), decrease in lymphocyte count (OR 3.57, 95% CI 2 to 6.67; RD 17.1%, 95% CI 7.5% to 30.7%), high aspartate aminotransferase (AST) (OR 3.5, 95% CI 1.59–7.71; RD 11.1%, 95% CI 4% to 16.8%), albumin increase (OR 1.53, 95% CI 1.32 to 1.78; RD 4.2%, 95% CI 2.5% to 6%) and creatinine increase (OR 1.14, 95%CI 1.02 to 1.28; RD 1.1%, 95% CI 0.2% to 2.3%).

**Others.** SOFA score> 2 (OR 1.97, 95% CI 1.22 to 3.2; RD 7.3%, 95% CI 1.8% to 15%).

**Table 1. Prognostic factors for mortality and/or severe COVID-19 disease.**

| Prognostic factor | Mortality | | | | | Severe COVID-19 disease | | | | |
|---|---|---|---|---|---|---|---|---|---|---|
| | Number of patients (studies) | Odds ratio (95%CI) | Risk without prognostic factor | Risk with prognostic factor | Certainty of the evidence | Number of patients (studies) | Odds ratio (95%CI) | Risk without prognostic factor | Risk with prognostic factor | Certainty of the evidence |
| Socio-demographic characteristics | | | | | | | | | | |
| Age Definition: 10 years increase | 11962 (19) | 1.80 (1.54–2.10) | 9% | 15.1% | ⊕⊕⊕⊕ HIGH | 14456 (53) | 1.63 (1.47–1.80) | 13% | 19.6% | ⊕⊕⊕⊕ HIGH |
| | | | 6.1% increase in mortality. Between 4.2% more and 8.2% more | | | | | 6.6% increase in severe COVID-19 disease. Between 5% more and 8.2% more | | |
| Sex Definition: Male | 31948 (58) | 1.72 (1.5–1.98) | 8% | 13% | ⊕⊕⊕◯ MODERATE[d] | 25032 (122) | 1.53 (1.4–1.67) | 10.8% | 15.5% | ⊕⊕⊕⊕ HIGH |
| | | | 5% increase in mortality. Between 4% more and 7% more | | | | | 4.7% increase in severe COVID-19 disease. Between 3.7% more and 5.6% more | | |
| Smoking Definition: Active, present smoker | 12025 (16) | 1.57 (1.19–2.07) | 8.7% | 13% | ⊕⊕⊕⊕ HIGH | 9147 (45) | 1.65 (1.25–2.17) | 12.1% | 18.4% | ⊕⊕⊕◯ MODERATE[d] |
| | | | 4.3% increase in mortality. Between 1.5% more and 7.5% more | | | | | 6.3% increase in severe COVID-19 disease. Between 2.7% more and 10.2% more | | |
| Medical illness and patient history | | | | | | | | | | |
| Any chronic condition or comorbidities | 4406 (16) | 3.3 (2.18–5) | 5.9% | 16.2% | ⊕⊕⊕◯ MODERATE[a] | 6640 (40) | 3.16 (2.71–3.68) | 8.2% | 20.1% | ⊕⊕⊕⊕ HIGH |
| | | | 10.3% increase in mortality. Between 6.8% more and 13.4% more | | | | | 12% increase in severe COVID-19 disease. Between 10.6% more and 13.2% more | | |
| Cerebrovascular disease Definition: History of stroke or CNS disease | 15294 (26) | 2.85 (2.02–4.01) | 8.7% | 21.3% | ⊕⊕⊕⊕ HIGH | 11050 (42) | 2.67 (1.84–3.87) | 12.7% | 27.8% | ⊕⊕⊕◯ MODERATE[d] |
| | | | 12.6% increase in mortality. Between 7.5% more and 18.5% more | | | | | 15.1% increase in severe COVID-19 disease. Between 8.4% more and 22.8% more | | |
| COPD | 34759 (41) | 2.43 (1.88–3.14) | 8.5% | 18.4% | ⊕⊕⊕⊕ HIGH | 15468 (65) | 2.7 (2.14–3.4) | 12.6% | 27.9% | ⊕⊕⊕⊕ HIGH |
| | | | 9.8% increase in mortality. Between 6.4% more and 13.6% more | | | | | 15.3% increase in severe COVID-19 disease. Between 11% more and 20% more | | |
| Chronic kidney disease Definition: KDIGO definition of CKD | 23448 (28) | 2.27 (1.69–3.05) | 8.5% | 17.2% | ⊕⊕⊕⊕ HIGH | 12056 (42) | 2.21 (1.51–3.24) | 12.8% | 24.5% | ⊕⊕◯◯ LOW[a,d] |
| | | | 8.8% increase in mortality. Between 5.1% more and 12.9% more | | | | | 11.7% increase in severe COVID-19 disease. Between 5.4% more and 19.2% more | | |
| Cardiovascular disease Definition: Coronary heart disease or congestive heart failure | 37156 (51) | 2.12 (1.77–2.56) | 8.1% | 15.5% | ⊕⊕⊕◯ MODERATE[d] | 16679 (73) | 3.34 (2.71–4.1) | 12.2% | 31.3% | ⊕⊕⊕◯ MODERATE[d] |
| | | | 7.5% increase in mortality. Between 5.4% more and 9.7% more | | | | | 19.1% increase in severe COVID-19 disease. Between 15.1% more and 23.1% more | | |
| Cardiac arrhythmia | 12729 (6) | 2.13 (1.72–2.65) | 7% | 13.6% | ⊕⊕⊕⊕ HIGH | 747 (4) | 16.51 (6.69–40.77) | 6.5% | 35.5% | ⊕⊕◯◯ LOW[a,c,e] |
| | | | 6.5% increase in mortality. Between 4.7% more and 8.4% more | | | | | 29% increase in severe COVID-19 disease. Between 22.6% more and 32.3% more | | |

(*Continued*)

**Table 1.** (Continued)

| Prognostic factor | Mortality | | | | | Severe COVID-19 disease | | | | |
|---|---|---|---|---|---|---|---|---|---|---|
| | Number of patients (studies) | Odds ratio (95%CI) | Risk without prognostic factor | Risk with prognostic factor | Certainty of the evidence | Number of patients (studies) | Odds ratio (95%CI) | Risk without prognostic factor | Risk with prognostic factor | Certainty of the evidence |
| Arterial hypertension | 31341 (52) | 2.02 (1.71–2.38) | 7% | 13% | ⊕⊕⊕⊕ HIGH | 20817 (94) | 2.5 (2.21–2.92) | 11.1% | 23.3% | ⊕⊕⊕○ MODERATE[d] |
| | | | 6% increase in mortality. Between 4.5% more and 7.3% more | | | | | 12.1% increase in severe COVID-19 disease. Between 10.4% more and 14.4% more | | |
| Diabetes | 30303 (52) | 1.84 (1.61–2.1) | 7.9% | 13.6% | ⊕⊕⊕⊕ HIGH | 21381 (97) | 2.51 (2.2–2.87) | 12% | 25.2% | ⊕⊕⊕⊕ HIGH |
| | | | 5.6% increase in mortality. Between 4.3% more and 7% more | | | | | 13.2% increase in severe COVID-19 disease. Between 11% more and 15.5% more | | |
| Dementia | 8922 (3) | 1.54 (1.31–1.81) | 9% | 13.2% | ⊕⊕⊕⊕ HIGH | 0 (0) | NA | NA | NA | NA |
| | | | 4.2% increase in mortality. Between 2.5% more and 6.2% more | | | | | NA | | |
| Obesity: BMI > 25–30 | 9127 (3) | 1.41 (1.15–1.74) | 8.5% | 11.5% | ⊕⊕⊕⊕ HIGH | 1140 (8) | 3.74 (2.37–5.89) | 10.2% | 35% | ⊕⊕⊕⊕ HIGH |
| | | | 3.1% increase in mortality. Between 1.2% more and 5.1% more | | | | | 16.7% increase in severe COVID-19 disease. Between 9.6% more and 24.7% more | | |
| Cancer Definition: Solid or active haematologic cancer | 22734 (25) | 1.35 (1.17–1.55) | 8.9% | 11.6% | ⊕⊕⊕⊕ HIGH | 15156 (58) | 2.06 (1.64–2.58) | 12.8% | 23.2% | ⊕⊕⊕○ MODERATE[d] |
| | | | 2.7% increase in mortality. Between 1.4% more and 4.2% more | | | | | 10.4% increase in severe COVID-19 disease. Between 6.6% more and 14.5% more | | |
| Dyslipidemia | 11273 (4) | 1.26 (1.06–1.5) | 8.9% | 11% | ⊕⊕⊕○ MODERATE[b] | 559 (4) | 0.63 (0.22–1.83) | 13.1% | 8.7% | ⊕⊕○○ LOW [a,b] |
| | | | 2.1% increase in mortality. Between 0.5% more and 3.9% more | | | | | 4.4% decrease in severe COVID-19 disease. Between 10% less and 8.3% more | | |
| Symptoms, vital signs and physical examination | | | | | | | | | | |
| Respiratory failure Definition: increased respiratory rate, abnormal blood gases (hypoxemia, hypercapnia, or both), and evidence of increased work of breathing | 1887 (8) | 21.17 (4.9–91.3) | 3.1% | 23.4% | ⊕⊕⊕○ MODERATE[a] | 1156 (7) | 23.21 (12.07–44.62) | NA | NA | NA |
| | | | 20.3% increase in mortality. Between 13.4% more and 22.4% more | | | | | NA | | |
| Low blood pressure Definition SBP less than 90–100 mmHg | 1269 (2) | 6.7 (3.14–14.33) | 9% | 39.9% | ⊕⊕⊕○ MODERATE[a] | 480 (2) | 1.29 (0.72–2.29) | NA | NA | NA |
| | | | 30.9% increase in mortality. Between 14.7% more and 49.6% more | | | | | NA | | |
| Hypoxemia Definition: Low digital saturation (below 90–93%) | 1047 (5) | 5.46 (2.05–14.53) | 2.3% | 9.1% | ⊕⊕⊕○ MODERATE[a] | 1331 (5) | 4.69 (1.56–14.09) | NA | NA | NA |
| | | | 6.7% increase in mortality. Between 4.2% more and 7.7% more | | | | | NA | | |

(*Continued*)

**Table 1.** (Continued)

| Prognostic factor | Mortality | | | | | Severe COVID-19 disease | | | | |
|---|---|---|---|---|---|---|---|---|---|---|
| | Number of patients (studies) | Odds ratio (95%CI) | Risk without prognostic factor | Risk with prognostic factor | Certainty of the evidence | Number of patients (studies) | Odds ratio (95%CI) | Risk without prognostic factor | Risk with prognostic factor | Certainty of the evidence |
| Tachycardia Definition: More than 90–100 bpm | 1269 (2) | 2.61 (1.62–4.22) | 9% | 20.5% | ⊕⊕⊕○ MODERATE[a] | 78 (1) | 1.54 (0.31–7.58) | 13% | 18.7% | ⊕○○○ VERY LOW[a,f] |
| | | | 11.5% increase in mortality. Between 4.8% more and 20.4% more | | | | | 5.7% increase in severe COVID-19 disease. Between 8.6% less and 40.1% more | | |
| Dyspnea Definition: Dyspnea or shortness of breath | 6613 (28) | 3.45 (2.72–4.38) | 4.9% | 13.8% | ⊕⊕⊕⊕ HIGH | 16803 (78) | 4.23 (3.32–5.38) | 9.3% | 27.8% | ⊕⊕○○ LOW[a,d] |
| | | | 8.9% increase in mortality. Between 7.5% more and 10.2% more | | | | | 18.5% increase in severe COVID-19 disease. Between 15.4% more and 21.3% more | | |
| Anorexia | 1483 (8) | 2.16 (1.14–4.12) | 7.3% | 14.4% | ⊕⊕⊕○ MODERATE[c] | 5495 (26) | 2.86 (2.16–3.84) | 10.4% | 24% | ⊕⊕⊕○ MODERATE[a] |
| | | | 7% increase in mortality. Between 1.1% more and 13.1% more | | | | | 13.6% increase in severe COVID-19 disease. Between 9.8% more and 17.5% more | | |
| Tachypnea Definition: More than 20–24 bpm | 202 (1) | 1.21 (1.12–1.31) | 7.6% | 9% | ⊕⊕⊕○ MODERATE[a] | 518 (7) | 7.51 (1.66–33.91) | 13% | 52.9% | ⊕⊕⊕○ MODERATE[d] |
| | | | 1.4% increase in mortality. Between 0.9% more and 1.9% more | | | | | 39.9% increase in severe COVID-19 disease. Between 6.9% more and 70.5% more | | |
| Haemoptysis | 781 (5) | 2.91 (0.74–11.4) | 8.3% | 20.6% | ⊕⊕○○ LOW[a,b] | 3317 (14) | 4.39 (2.18–8.81) | 12.7% | 38.6% | ⊕⊕⊕○ MODERATE[a] |
| | | | 12.3% increase in mortality. Between 2.2% less and 34.6% more | | | | | 25.9% increase in severe COVID-19 disease. Between 11.4% more and 42.1% more | | |
| Abdominal pain | 1127 (5) | 1.06 (0.53–2.11) | 9% | 9.5% | ⊕⊕○○ LOW[a,b] | 4896 (22) | 1.95 (1.36–2.79) | 12.7% | 22% | ⊕⊕⊕○ MODERATE[a] |
| | | | 5% increase in mortality. Between 4% less and 8% more | | | | | 9.4% increase in severe COVID-19 disease. Between 4% more and 15.8% more | | |
| Fatigue | 3725 (21) | 1.66 (1.27–2.17) | 7.1% | 11.1% | ⊕⊕○○ LOW[a,d] | 13262 (71) | 1.41 (1.19–1.68) | 11.6% | 15.5% | ⊕⊕⊕○ MODERATE[d] |
| | | | 4% increase in mortality. Between 1.9% more and 5.9% more. | | | | | 3.9% increase in severe COVID-19 disease. Between 2% more and 5.9% more. | | |
| Fever. Definition: More than 37.5˚C | 6154 (31) | 1.04 (0.77–1.4) | 8.7% | 9.1% | ⊕⊕○○ LOW[a,b] | 20026 (102) | 1.84 (1.54–2.21) | 9.3% | 15.4% | ⊕⊕⊕○ MODERATE[d] |
| | | | 0.3% increase in mortality. Between 2.3% less and 2.5% more | | | | | 6.1% increase in severe COVID-19 disease. Between 4.5% more and 7.6% more | | |
| Myalgia/arthralgia Definition: myalgia or arthralgias | 3436 (18) | 0.96 (0.77–1.23) | 9.1% | 8.7% | ⊕⊕○○ LOW[a,b] | 13814 (61) | 1.29 (1.03–1.61) | 12.5% | 15.6% | ⊕⊕⊕○ MODERATE[d] |
| | | | 0.3% decrease in mortality. Between 2% less and 1.8% more | | | | | 3% increase in severe COVID-19 disease. Between 0.3% more and 5.9% more | | |
| Laboratory measures (blood or plasma) | | | | | | | | | | |
| High procalcitonin Definition: More than 0.01–05 ng/ml | 4735 (10) | 12.42 (7.18–21.5) | 6.3% | 38.5% | ⊕⊕⊕○ MODERATE[c] | 7923 (28) | 5.13 (3.16–8.35) | 10.6% | 35.4% | ⊕⊕○○ LOW[a,d] |
| | | | 32.3% increase in mortality. Between 25% more and 38.1% more | | | | | 24.8% increase in severe COVID-19 disease. Between 16.7% more and 32.3% more | | |

*(Continued)*

**Table 1.** (Continued)

| Prognostic factor | Mortality | | | | | Severe COVID-19 disease | | | | |
|---|---|---|---|---|---|---|---|---|---|---|
| | Number of patients (studies) | Odds ratio (95%CI) | Risk without prognostic factor | Risk with prognostic factor | Certainty of the evidence | Number of patients (studies) | Odds ratio (95%CI) | Risk without prognostic factor | Risk with prognostic factor | Certainty of the evidence |
| Myocardial injury Definition: Reported as myocardial injury or as increase in blood troponins | 3855 (21) | 10.89 (5.39–22.04) | 3.5% | 20.4% | ⊕⊕⊕○ MODERATE[d] | 3627 (20) | 10 (6.84–14.62) | 11.1% | 51.3% | ⊕⊕⊕⊕ HIGH |
| | | | 16.9% increase in mortality. Between 13.4% more and 19% more | | | | | 40.2% increase in severe COVID-19 disease. Between 33.1% more and 46.4% more | | |
| High WBC Definition: greater than 10.0 x 10⁹/L | 2870 (10) | 4.06 (2.7–6.12) | 7.8% | 24.7% | ⊕⊕⊕○ MODERATE[d] | 9331 (32) | 4.67 (3.17–6.88) | 11.2% | 35.6% | ⊕⊕⊕⊕ HIGH |
| | | | 16.9% increase in mortality. Between 11% more and 23.3% more | | | | | 24.3% increase in severe COVID-19 disease. Between 17.3% more and 31.2% more | | |
| High lactate: Definition More than 1.5–2.2 mmol/L | 1078 (1) | 3.66 (2.26–5.94) | 7.3% | 21.7% | ⊕⊕⊕○ MODERATE[a] | 812 (3) | 3.74 (0.69–20.16) | 12.1% | 33.3% | ⊕○○○ VERY LOW [a,b,d] |
| | | | 14.3% increase in mortality. Between 8.3% more and 20.6% more | | | | | 21.2% increase in severe COVID-19 disease. Between 3.7% less and 51.4% more | | |
| Low platelet count Definition: Less than 100–150 x 10⁹/L | 3676 (10) | 5.43 (2.55–11.56) | 5% | 19.3% | ⊕⊕⊕⊕ HIGH | 8081 (32) | 1.93 (1.52–2.46) | 11.1% | 19.2% | ⊕⊕○○ LOW[a,d] |
| | | | 14.3% increase in mortality. Between 8.3% more and 18.6% more | | | | | 8% increase in severe COVID-19 disease. Between 5% more and 11.1% more | | |
| High D-dimer Definition: More than 500–1000 ng/ml | 4361 (17) | 4.81 (3.15–7.34) | 4.3% | 15.6% | ⊕⊕⊕○ MODERATE[d] | 6356 (24) | 3.27 (2.46–4.36) | 8.2% | 20.7% | ⊕⊕⊕○ MODERATE[d] |
| | | | 11.2% increase in mortality. Between 8.8% more and 13.1% more | | | | | 12.5% increase in severe COVID-19 disease. Between 9.8% more and 14.8% more | | |
| High LDH Definition: More than 240–250 U/L | 1440 (6) | 4.09 (1.18–14.17) | 4.7% | 15.2% | ⊕⊕⊕○ MODERATE[d] | 7955 (26) | 4.48 (3.21–6.25) | 7.8% | 23.9% | ⊕⊕⊕○ MODERATE[d] |
| | | | 10.4% increase in mortality. Between 1.4% more and 15.3% more | | | | | 16.2% increase in severe COVID-19 disease. Between 13.1% more and 18.8% more | | |
| High CRP Definition: More than 1–100 mg/l | 2107 (8) | 6.6 (3.36–12.99) | 2.3% | 10.3% | ⊕⊕⊕○ MODERATE[d] | 9094 (37) | 4.5 (3.1–6.23) | 6.3% | 19.5% | ⊕⊕⊕⊕ HIGH |
| | | | 7.9% increase in mortality. Between 6.4% more and 8.7% more | | | | | 13.2% increase in severe COVID-19 disease. Between 10.8% more and 14.9% more | | |
| Decrease in Lymphocyte count Definition: per 1 x 10⁹ U/L decrease | 544(3) | 3.57 (2–6.67) | 9% | 26.1% | ⊕⊕⊕○ MODERATE[d] | 1909 (7) | 2.28 (1.21–4.30) | 13% | 25.4% | ⊕⊕⊕○ MODERATE[d] |
| | | | 17.1% increase in mortality. Between 7.5% more and 30.7% more | | | | | 12.4% increase in severe COVID-19 disease. Between 2.3% more and 26.1% more | | |
| High AST level Definition: More than 32–40 U/l | 2969 (7) | 3.5 (1.59–7.71) | 6% | 17.1% | ⊕⊕⊕○ MODERATE[d] | 9179 (32) | 3.41 (2.7–4.3) | 9.9% | 25.8% | ⊕⊕⊕○ MODERATE[a] |
| | | | 11.1% increase in mortality. Between 4% more and 16.8% more | | | | | 15.8% increase in severe COVID-19 disease. Between 12.7% more and 18.8% more | | |
| Decrease in albumin: Definition: 20 g/L decrease | 336 (3) | 1.53 (1.32–1.78) | 9% | 13.2% | ⊕⊕⊕○ MODERATE[c] | 1266 (5) | 1.11 (1.01–1.21) | 13% | 14.2% | ⊕⊕⊕○ MODERATE[b] |
| | | | 4.2% increase in mortality. Between 2.5% more and 6% more | | | | | 1.2% increase in severe COVID-19 disease. Between 0.1% more and 2.3% more | | |

(*Continued*)

**Table 1.** (Continued)

| Prognostic factor | Mortality | | | | | Severe COVID-19 disease | | | | |
|---|---|---|---|---|---|---|---|---|---|---|
| | Number of patients (studies) | Odds ratio (95%CI) | Risk without prognostic factor | Risk with prognostic factor | Certainty of the evidence | Number of patients (studies) | Odds ratio (95%CI) | Risk without prognostic factor | Risk with prognostic factor | Certainty of the evidence |
| Increase in creatinine Definition: per 1 mg/dL increase | 1508 (9) | 1.14 (1.02–1.28) | 9% | 10.1% | ⊕⊕⊕○ MODERATE[b] | 1116 (4) | 1.89 (0.87–4.10) | 13% | 22% | ⊕⊕⊕○ MODERATE[b] |
| | | | 1.1% increase in mortality. Between 0.2% more and 2.3% more | | | | | 9% increase in severe COVID-19 disease. Between 1.5% less and 25% more | | |
| High Neutrophil count Definition: greater than 6.3 x 10⁹/L | 727 (2) | 6.78 (3.07–14.97) | 5.2% | 23% | ⊕⊕○○ LOW[a,c] | 4945 (16) | 5.66 (3.71–8.63) | 9% | 31% | ⊕⊕⊕○ MODERATE[a] |
| | | | 17.8% increase in mortality. Between 10% more and 23% more | | | | | 22% increase in severe COVID-19 disease. Between 17% more and 27% more | | |
| High BNP: More than 500–900 pg/mL | 1283 (6) | 3.27 (1.24–8.63) | 7% | 19% | ⊕⊕○○ LOW[a,d] | 1086 (1) | 4.99 (3.2–7.77) | 9.4% | 30.9% | ⊕⊕⊕○ MODERATE[a] |
| | | | 12% increase in mortality. Between 1.9% more and 21.9% more | | | | | 21.5% increase in severe COVID-19 disease. Between 15.5% more and 26.7% more | | |
| High BUN Definition: mmol/L, > 5.2–9.5 | 1258 (2) | 10.56 (6.76–16.48) | 5.2% | 29.6% | ⊕⊕○○ LOW[a,c,e] | 3890 (10) | 3.66 (2.82–4.74) | 11.1% | 30.2% | ⊕⊕⊕○ MODERATE[a] |
| | | | 24.4% increase in mortality. Between 20.2% more and 27.7% more | | | | | 19.1% increase in severe COVID-19 disease. Between 14.8% more and 23.4% more | | |
| High CPK Definition: More than 185–200 U/L | 407 (3) | 1.35 (0.58–3.14) | 8.8% | 11.5% | ⊕⊕○○ LOW[a,b] | 3292 (13) | 3.1 (2.32–4.16) | 11.5% | 28.1% | ⊕⊕⊕○ MODERATE[a] |
| | | | 2.7% increase in mortality. Between 3.7% less and 13% more | | | | | 16.5% increase in severe COVID-19 disease. Between 11.7% more and 21.6% more | | |
| High total bilirubin Definition: More than 17–21 pg/ml | 2715 (3) | 3.03 (1.87–4.92) | 8.1% | 20.7% | ⊕⊕○○ LOW[a,c] | 5098 (14) | 2.94 (2.18–3.97) | 12.5% | 29.3% | ⊕⊕⊕○ MODERATE[a] |
| | | | 12.6% increase in mortality. Between 6.3% more and 19.9% more | | | | | 16.8% increase in severe COVID-19 disease. Between 11.3% more and 22.9% more | | |
| High interleukin-6 Definition: More than 5–20 pg/ml | 436 (4) | 1.31 (0.14–12.27) | 8.1% | 10.3% | ⊕⊕○○ LOW[a,b] | 1211 (7) | 7.36 (2.97–18.27) | 6.5% | 26.2% | ⊕⊕⊕○ MODERATE[a] |
| | | | 2.2% increase in mortality. Between 11.6% less and 15% more | | | | | 19.7% increase severe COVID-19 disease. Between 12.2% more and 23.9% more | | |
| High Definition More than 10–20 mm/H | 628 (3) | 0.89 (0.54–1.45) | 9.7% | 8.7% | ⊕⊕○○ LOW[a,b] | 2557 (12) | 3.08 (2.04–4.65) | 6.6% | 15.6% | ⊕⊕⊕○ MODERATE[a] |
| | | | 1% decrease in mortality. Between 5.6% less and 2.8% more | | | | | 9.4% increase in severe COVID-19 disease. Between 6.7% more and 11.3% more | | |
| Radiological signs | | | | | | | | | | |
| Pleural effusion Definition: X ray or CT assessment | 820 (5) | 1.38 (0.63–3.06) | 8.8% | 11.7% | ⊕⊕○○ LOW[a,b] | 5289 (23) | 3.31 (2.03–5.38) | 12.5% | 32% | ⊕⊕⊕○ MODERATE[d] |
| | | | 3% increase in mortality. Between 3% less and 13% more | | | | | 19% increase in severe COVID-19 disease. Between 10% more and 30% more | | |
| Consolidation pattern Definition: X ray or CT assessment | 795 (4) | 1.93 (1.31–2.84) | 7.5% | 13% | ⊕⊕○○ LOW[a,c] | 6133 (27) | 2.46 (1.54–3.93) | 11.2% | 23.2% | ⊕⊕⊕○ MODERATE[d] |
| | | | 5.8% increase in mortality. Between 2.3% more and 9.4% more | | | | | 12% increase in severe COVID-19 disease. Between 5.4% more and 18.8% more | | |

(*Continued*)

**Table 1.** (Continued)

| Prognostic factor | Mortality | | | | | Severe COVID-19 disease | | | | |
|---|---|---|---|---|---|---|---|---|---|---|
| | Number of patients (studies) | Odds ratio (95%CI) | Risk without prognostic factor | Risk with prognostic factor | Certainty of the evidence | Number of patients (studies) | Odds ratio (95%CI) | Risk without prognostic factor | Risk with prognostic factor | Certainty of the evidence |
| Others | | | | | | | | | | |
| High SOFA score Definition: More than 2 | 585 (3) | 1.97 (1.22–3.2) | 9% | 16.3% | ⊕⊕⊕○ MODERATE[d] | 92 (2) | 21.31 (6.26–72.6) | 13% | 76.1% | ⊕⊕○○ LOW[a,c] |
| | | | 7.3% increase in mortality. Between 1.8% more and 15% more | | | | | 63% increase in severe COVID-19 disease. Between 35.3% more and 78.6% more | | |

Glossary

FDP: Fibrin Degradation Product.

PT: Prothrombin time.

APTT: Activated partial thromboplastin time.

APACHE: Acute Physiology And Chronic Health Evaluation II.

SOFA: The sequential organ failure assessment score.

qSOFA: Quick sepsis related organ failure assessment.

AST: Aspartate aminotransferase.

ALT: Alanine aminotransferase.

BUN: Blood urea nitrogen.

NA: Not applicable, either because there is no information or because the addressed variable does not represent a potential prognostic factor in that clinical scenario.

Explanations

a. Risk of bias due to study limitations (unadjusted estimates, inappropriate prognostic factor or outcome assessment, inappropriate population inclusion criteria, study attrition).

b. Imprecision: Confidence interval includes significant and non-significant risk increase.

c. Imprecision due to fragility: Less than 200 events.

d. Inconsistency: Unexplained visual heterogeneity.

e. Risk of selective reporting: Most of the pooled estimate weight from studies that performed multivariable analysis but did not report adjusted estimates.

f. Very serious imprecision: Very wide confidence interval.

## Prognostic factors for severe COVID-19 disease

We investigated 96 candidate prognostic factors for severe COVID-19 disease from 131 studies including 28538 patients. S3 Table provides a summary of findings for all the candidate prognostic factors and S2 Appendix includes the corresponding forest plots.

In addition to identified prognostic factors for mortality, we found high or moderate certainty that the following 14 variables provide valuable prognostic information on severe COVID-19 disease outcome (Table 1).

**Symptoms, vital signs and physical examination factors.** Haemoptysis (OR 4.39, 95% CI 2.18 to 8.81; RD 25.9%, 95% CI 11.4% to 42.1%), abdominal pain (OR 1.95, 95% CI 1.36 to 1.79; RD 9.4%, 95% CI 4% to 15.8%), fatigue (OR 1.41, 95% CI 1.19 to 1.68; RD 3.9%, 95% CI 2% to 5.9%), fever (OR 1.84, 95% CI 1.54 to 2.21; RD 6.1%, 95% CI 4.5% to 7.6%) and myalgia or arthralgia (OR 1.29, 95% CI 1.03 to 1.61; RD 3%, 95%CI 0.3% to 5.9%).

**Laboratory factors (measured in blood or plasma).** High neutrophil count (OR 5.66, 95% CI 3.71 to 8.63; RD 22%, 95% CI 17% to 27%), high B-type natriuretic peptide (BNP) (OR 4.99, 95% CI 3.2 to 7.77; RD 21.5%, 95% CI 15.5% to 26.7%), High urea nitrogen (BUN) (OR 3.66, 95% CI 2.82 to 4.74; RD 19.1%, 95% CI 14.8% to 23.4%), high creatine kinase (CK) (OR 3.1, 95% CI 2.32 to 4.16; RD 16.5%, 95% CI 11.7% to 21.6%), high bilirubin (OR 2.94, 95% CI

2.18 to 3.97; RD 16.8%, 95% CI 11.3% to 22.9%), high interleukin-6 (IL-6) (OR 7.36, 95% CI 2.97 to 18.27; RD 13.3%, 95% CI 8.5% to 15.9%), high erythrocyte sedimentation rate (ESR) (OR 3.08, 95% CI 2.04 to 4.65; RD 9.4%, 95% CI 6.7% to 11.3%).

**Radiological factors.**   Consolidative infiltrate (OR 2.46, 95% CI 1.54 to 3.93; RD 12%, 95% CI 5.4% to 18.8%) and pleural effusion (OR 3.31, 95% CI 2.03 to 5.38; RD 19%, 95% CI 10% to 30%).

## Other analysed variables

The remaining variables analysed were: asthma, tuberculosis, HIV infection, immunocompromise, autoimmune disease, malnutrition, chronic liver disease, thyroid disease, chronic gastric disease, chest pain, high fever, cough, rhinorrhea, odynophagia, conjunctivitis, sputum production, enlarged lymph nodes, rash, headache, vomits, diarrhea, anemia, low WBC, low neutrophil count, glomerular filtration rate, blood urea, cystatin C, prothrombin time, APTT time, ferritin, cholinesterase, alanine aminotransferase (ALT), fibrinogen degradation products, globulin, prealbumin, blood glucose, alfa-HBDH, low density lipoprotein (LDL), triglycerides, any abnormal radiologic finding, radiological interstitial pattern, ground glass opacity, crazy paving pattern, radiological evidence of enlarged lymph nodes, bilateral radiological compromise, APACHE (Acute Physiology And Chronic Health Evaluation II), qSOFA (quick sepsis related organ failure assessment).

For all of these variables we found low or very low certainty evidence both for mortality and severe COVID-19 disease. Hence, it is uncertain if these variables provide prognostic value in the context of COVID-19 infected patients.

## Additional analysis

We performed a sensitivity analysis on mortality outcome using a non-contextualized approach and assuming adjusted estimates as at low risk of being biased (less demanding risk of bias approximation in comparison with the primary analysis) (see methods, risk of bias assessment in S1 Text). The results were similar to the primary analysis. However our certainty increased to moderate or high for some prognostic factors for which we had very low or low certainty: Chest pain, cough, sputum production, anemia, high ferritin, high ALT, increase in blood glucose and high APACHE score.

A second sensitivity analysis in which we set a significantly higher baseline mortality risk (26%) for patients with severe but non-critical COVID-19 disease did not show differences with the primary analysis.

When we found inconsistent results for severe COVID-19 disease outcome (n = 43), we performed subgroup analyses accounting on outcome definition. Observed heterogeneity could not be explained by this analysis for any of the candidate prognostic factors explored (see S3 Appendix).

## Discussion

In this systematic review we evaluated prognostic factors for poor outcome in patients with covid-19 infectious disease. We found 49 variables that provide valuable prognostic information for mortality and/or severe COVID-19 disease. Identified prognostic factors include socio-demographic characteristics (age, male sex and smoking) medical illness and patients history information (comorbidities including chronic respiratory, cardiac and endocrinologic conditions), physical examination findings (respiratory failure related symptoms as well as general clinical condition deterioration), laboratory (multiple biomarkers and alterations in basic laboratory tests) and radiological findings (consolidation pattern and pleural effusion)

([Table 1]). Overall the risk of severe COVID-19 disease or death resulted higher in older patients and those with previous medical conditions including COPD and cardiovascular disease as some of the most relevant predictors. Additionally, those patients that presented with clinical signs and symptoms suggesting respiratory failure or laboratory biomarkers showing inflammation or organ damage were also at increased risk of severe COVID-19 disease or death. Radiological features did not show good predictive value.

## Strengths and limitations of the study

Our systematic review has a number of strengths. First, it provides the most comprehensive and trustworthy body of evidence up to date as it includes a significant number of studies not included in prior reviews. Secondly, we followed the GRADE approach to summarize and rate the certainty on the evidence. And thirdly, we presented our results both as relative estimates of association as well as absolute risk differences and used the latter to interpret and analyse our results. We consider that the absolute risk modification that can be attributed to a prognostic factor is a critical piece of information for those aiming to make decisions using prognostic information.

Regarding limitations, most of the studies included in this review were not published in peer review journals (only as preprint) at the time we performed the search. We identified most of those studies by cross reference search in google scholar, but it is possible that some may have not been detected by our search strategy. Additionally, given the high publication speed of COVID-19 studies it is probable that new relevant information not included in our review is available at the time our review is published. We aim to address this issue by updating our results in the short term. Although we made efforts to identify data duplication, in many instances it was not clear if studies reported, totally or partially, on the same cohorts of patients hence we assume there is a considerable chance of some degree of data overlap between included publications.

Significant variability in study design, study type, patient eligibility criteria, prognostic factor definition was observed, however, given the huge amount of information analysed, it was not feasible to explore subgroup effects accounting for those differences.

In analysing our results we implemented a minimally contextualized approach for which we arbitrarily set thresholds to define the minimal important risk difference necessary to assume valuable prognostic information. As the degree of contextualization was minimal, we set very low thresholds (near the point of no effect). We acknowledge that readers might find those thresholds inappropriate, hence we also provided relative estimates of association which can be used with alternative thresholds or analytical approximations (e.g partially contextualized approach). In addition, for some candidate prognostic factors, baseline risk could not be adjusted for prevalence which might have resulted in an overestimation or underestimation of the risk difference estimates. Finally, for risk of bias assessment, we defined a set of requirements for dealing with potential confounders that were not previously validated.

## Relation to prior work

We identified multiple systematic reviews addressing prognostic factors in patients with COVID-19 infectious disease [232–251]. All analysed certain prognostic factors or groups or prognostic factors that we included in the present review, and measured mortality and/or disease severity as outcomes. Most of the reported results are in consonance with our findings with only a few exceptions. Kumar et.al [246] reported diarrhea, productive cough and high ALT as prognostic factors however we found low certainty evidence on those variables in our primary analysis. Wang et.al [249] reported no association between chronic kidney disease

and malignancy with poor outcomes in COVID-19 patients however we found that both conditions are associated with an increased risk of mortality and severe COVID-19 disease. Other significant differences of our review in relation to these prior reviews include multiple characteristics that were previously not identified as prognostic factors. In contrast to previous reviews, here we provide both relative and absolute estimates of risk and provide our certainty in those estimates.

Significant information has been published since our search was finalized. An update of the ISARIC registry [252] including 15194 hospitalised patients discharged or dead, the openSA-FELY registry [253] included 17425445 adults potentially exposed to COVID-19 infection and a Chinese registry [254] that included 44672 patients with COVID-19.

These studies identified the following variables as prognostic factors for COVID-19 related mortality: Age, sex (male), obesity, cardiovascular disease, diabetes, arterial hypertension, dyslipidemia, COPD, smoking, malignancy, cerebrovascular disease, dementia and chronic kidney disease. All these variables were captured by our analysis as predictors of COVID-19 related mortality or severe COVID-19 disease for which moderate or high certainty evidence exists. In addition, other prognostic factors were identified by these studies: race (not white) [253] and deprivation [253], two variables we did not explore, and immunocompromise [253], asthma [253], autoimmune diseases [253], and chronic liver disease [252,253], four variables for which we found low certainty evidence.

## Implications of study

Our approach considered two clinical scenarios in which we assumed that the prognostic information of each predictor for each outcome could potentially impact decision-making. In patients presenting with mild disease, predicting the risk of progression to severe status could support decisions on the level of healthcare required and more extensive follow up strategies. In the same way, in patients presenting with severe disease, predicting mortality risk could support the use of certain, more aggressive, therapeutic interventions. Clinicians or decision-makers can use our results to tailor management strategies for patients with COVID-19. For example, they could select a set of prognostic factors for which there is high certainty in a significant risk incremental increase (e.g Age, gender, comorbidities, respiratory failure and myocardial injury) and use them to define hospitalization rules for patients consulting to the emergency department. However, to what extent accounting for these prognostic factors will improve clinically important outcomes is a question that cannot be addressed with our results. Furthermore using information on multiple individual prognostic factors for outcome prediction is challenging. Multivariable models provide a solution to this limitation, however, considering the high demand for accurate risk prediction models for patients with COVID-19 [7], our work can also provide solid grounds for development of these prognostic tools.

## Conclusions

We have identified a set of variables that provide valuable prognostic information in patients with COVID-19 infectious disease. Clinicians and policy makers can use our results to tailor management strategies for patients with this condition while researchers can utilise our findings to develop multivariable prognostic models that could eventually facilitate decision-making and improve patient important outcomes.

## Supporting information

**S1 Table. Included studies characteristics.** This table presents detailed information on individual included studies.
(PDF)

**S2 Table. Risk of Bias (RoB) assessment.** This table contains a detailed RoB assessment of included studies.
(PDF)

**S3 Table. Summary of findings of all candidate variables.** This table presents the complete results including all assessed candidate variables.
(DOCX)

**S1 Text. Supplementary methods.** This file contains additional details on methods.
(DOCX)

**S1 File. PRISMA document.** This file contains the references of the pages according to PRISMA Statement.
(DOCX)

**S1 Appendix. Mortality forest plots.** This file contains the Forest plots for all assessed candidate variables.
(PDF)

**S2 Appendix. COVID-19 severe disease forest plots.** This file contains the Forest plots for all assessed candidate variables.
(PDF)

**S3 Appendix. Subgroup analyses forest plots.** This file contains the Forest plots for all performed subgroup analyses.
(PDF)

## Author Contributions

**Conceptualization:** Ariel Izcovich, Martín Alberto Ragusa.

**Data curation:** Ariel Izcovich, Martín Alberto Ragusa, Fernando Tortosa, María Andrea Lavena Marzio, Camila Agnoletti, Agustín Bengolea, Agustina Ceirano, Federico Espinosa, Ezequiel Saavedra, Verónica Sanguine, Alfredo Tassara, Candelaria Cid, Gabriel Rada.

**Formal analysis:** Ariel Izcovich, Martín Alberto Ragusa, Fernando Tortosa, Arnav Agarwal, Farid Foroutan.

**Investigation:** Ariel Izcovich, Martín Alberto Ragusa, Gabriel Rada.

**Methodology:** Ariel Izcovich, Martín Alberto Ragusa.

**Project administration:** Ariel Izcovich.

**Supervision:** Ariel Izcovich.

**Validation:** Ariel Izcovich, Martín Alberto Ragusa.

**Writing – original draft:** Ariel Izcovich, Martín Alberto Ragusa, Fernando Tortosa, Gabriel Rada.

**Writing – review & editing:** Ariel Izcovich, Martín Alberto Ragusa, Fernando Tortosa, María Andrea Lavena Marzio, Camila Agnoletti, Agustín Bengolea, Agustina Ceirano, Federico

Espinosa, Ezequiel Saavedra, Verónica Sanguine, Alfredo Tassara, Candelaria Cid, Hugo Norberto Catalano, Arnav Agarwal, Farid Foroutan, Gabriel Rada.

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
