## [Decision Letter · Decision Letter 0]

18 Sep 2020

PONE-D-20-20042

Prognostic factors for severity and mortality in patients infected with COVID-19: A systematic review

PLOS ONE

Dear Dr. Izcovich,

Thank you for submitting your manuscript to PLOS ONE. After careful consideration, we feel that it has merit but does not fully meet PLOS ONE’s publication criteria as it currently stands. Therefore, we invite you to submit a revised version of the manuscript that addresses the points raised during the review process.

We look forward to receiving your revised manuscript.

Kind regards,

Chiara Lazzeri

Academic Editor

PLOS ONE

Journal Requirements:

Reviewers' comments:

Reviewer's Responses to Questions

**Comments to the Author**

1. Is the manuscript technically sound, and do the data support the conclusions?

Reviewer #1: Yes

Reviewer #2: Yes

2. Has the statistical analysis been performed appropriately and rigorously? 

Reviewer #1: Yes

Reviewer #2: Yes

3. Have the authors made all data underlying the findings in their manuscript fully available?

Reviewer #1: Yes

Reviewer #2: Yes

4. Is the manuscript presented in an intelligible fashion and written in standard English?

Reviewer #1: Yes

Reviewer #2: Yes

5. Review Comments to the Author

Reviewer #1: First of all one has to applaud the authors for the sheer amount of data, they have provided in this systematic review! Generaly the manuscript is well writen, but still contains a few typing errors. Some formulations formulations are a matter of taste. Nevertheless, I would recommend to change them (detailed information can be found below).

Abstract:

Since you are describing/systematically reviewing prognostic factors, I would go so far and claim that it influences decision making! Prognostic factors are importnat as well, but they are not predictive factors. Therefore, developing a predictive model (would call it multivariate model not multivariable model) to estimate disease course, severity and mortality is, what the data support.

Since arterial hypertension, and cardiac arrythmias are mentioned seperately, it should be defined, what cardiovascular disease includes. Basically all laboratory test you mention are tested in the blood/serum or plasma. Please omit using blood ... (e.g. WBC instead of blood WBC). It is heavily affecting the fluidity of the text and adds no information. Propably it makes sense to define it at one point. Ultimately, it is also not consistant through the text/tables.

Covid-19 is defined as an infectious disease (does not include asymptomatic SARS-CoV-2 infections). Please change infection to infectious disease.

Introduction:

Please cite the RECOVERY trial (contains important and prospective reference information regarding mortality rate and the rate of severe Covid-19 disease). "Can may be used" (page 8) can be formulated a bit more eloquent (e.g. have to potential to be used/applied or can be potentially used). What is mentioned above about decision making applies also here.

Methods:

The sentence on page 17, chapter 1, line 3 is difficult to understand. "Exposed" is not the correct wording here. The biomarkers/clinical factors/symptoms/etc. are absent or present. It is also not clear if the biomarkers are measured/determined at baseline/SARS-CoV-2 diagnosis or crossed/never crossed the treshold during the complete disease until recovery. Please be more specific in your description. This is essential information!

On page 17, chapter 2, line 5 please change studies results to study results.

That you promote an update (AI based) is a huge plus point and might already be necessary!

The mortality has currently dropped to 5%. Your reference value is no longer up to date!!!

On page 13, line 2 please change "inseverty" to "in severity".

Although the cut-offs used are arbitrary, they seem to be weel chosen (e.g. 0.5% in overall mortality is one tenth of the current overall mortality).

Results:

Please change "Physical examination factors" to a more appropriate term. Fatique is rather subjective and a symptom. Personally I am also not aware of a test for myalgia. Rewording is required for example to "symptoms, vital signs & abnormal physical examination results".

Laboratory factors: as mentioned above.

Some terms (e.g. immuncompromise) need a better definition (lymphopenia, immunosupressive medication, etc. ???).

Please mention why you didn't look at co-medication! This might be the only factor, which is relevant for decision making!!!

Overall it is an important review, which summarizes the current biomarker landscape for Covid-19. The cut-off date (End of April 2020) is very early and new discoveries have been made since. NLR and lymphopenia have been even published earlier. They should actually have met the inclusion criteria, but are not included (Lancet, Lancet Resp. Med. Feb 2020). Data have been homgenised and adjusted. The tables, figures and supplementary figures are very informative and have been nicely prepared!

Results:

The mortality risks

Reviewer #2: This study sets out to report a very important issue: prognostic factors that may be used in decision-making related to the care of patients infected with COVID-19. Although, this topic is not novel, the current research investigated it in a much more detail way than previous studies. In addition, authors also raised the possible questions that must be tackled in near future in order to develop multivariable prognostic models that could eventually facilitate decision-making and improve patients’ outcomes.

Reviewer’s comments:

1. The definitions of severe COVID-19 disease are adopted from multiple studies and therefore are inconsistent. How do the authors eliminate the effect of inconsistent definitions on the association analysis?

2. When severe COVID-19 disease was not reported as an outcome, the authors considered ICU requirement, invasive mechanical ventilation (IVM) and acute respiratory distress syndrome (ARDS) as surrogate outcomes. Did authors analyze the correlation between primary outcome and surrogate outcomes in this study?

3. The ICU requirement, invasive mechanical ventilation (IVM) and acute respiratory distress syndrome (ARDS) were used as surrogate outcomes. As a results, the outcomes in this study include mortality, severe COVID-19 disease, ICU requirement, invasive mechanical ventilation (IVM) and acute respiratory distress syndrome (ARDS). Multiple outcomes make the results difficult to be interpretated. If those outcomes could be analyzed separately, the results can be interpretated more precisely.

4. In the analysis of multivariable models, the variables used for model adjustment include age, one comorbidity (e.g diabetes) and one parameter of disease severity (e.g. respiratory rate) at minimum. How do the authors validate that this setting is statistically appropriate?

5. In this study, the authors arbitrarily set thresholds to define important incremental increase in the risk of their outcomes, including mortality or severe COVID-19. How do the authors validate that those thresholds can select proper prognostic information?

6. The thresholds were set in 0.5% increase in mortality and 1% increase in severe COVID-19 disease. Do the authors analyze the correlation between 0.5% increase in mortality and 1% increase in severe COVID-19 disease?

7. The authors used the clinical scenario of a patient infected with COVID-19 with severe but not critical disease to assess the prognostic value on mortality. Please clarify the definition of critical disease.

8. When prevalence of prognostic factors was not available, the authors used described baseline risks (9% for mortality and 13 % for severe COVID-19 disease). This setting will inevitably lead to bias in the results.

9. This study enrolled 207 studies and only 7 were judged as low risk of bias as the remaining presented important limitations in at least one domain or item. Hence, there are multiple inevitable bias in the current study, limiting the reliability of the results.

10. There is too much information in Table 1. Is there any way to condense the information or split it into different tables?

11. The authors described that clinicians can use their results to tailor management strategies for patients with COVID-19. It would be of great help to the reader if the authors could formulate a summarized management strategy based on the results of this study.

6. PLOS authors have the option to publish the peer review history of their article (what does this mean?). If published, this will include your full peer review and any attached files.

Reviewer #1: No

Reviewer #2: No

---

## [Author Response · Author response to Decision Letter 0]

21 Oct 2020

We made several modifications to meet style requirements.

We now included all the forest plots in the appendices S5, S6 and S7

We included captions for supporting files as suggested.

4. Reviewer #1: First of all one has to applaud the authors for the sheer amount of data, they have provided in this systematic review! Generally the manuscript is well writen, but still contains a few typing errors. Some formulations formulations are a matter of taste. Nevertheless, I would recommend to change them (detailed information can be found below).

We want to thank the reviewers for taking the time and performing such a thorough review which will certainty result in significant improvements of our manuscript.

Abstract:

5. Since you are describing/systematically reviewing prognostic factors, I would go so far and claim that it influences decision making! Prognostic factors are importnat as well, but they are not predictive factors. Therefore, developing a predictive model (would call it multivariate model not multivariable model) to estimate disease course, severity and mortality is, what the data support.

We agree, but in the absence of well developed multivariate models clinicians and policymakers will need to rely on the best available evidence on individual prognostic factors, which we are providing in this review. We discuss this in page 23: “Implications of study”

6. Since arterial hypertension, and cardiac arrythmias are mentioned seperately, it should be defined, what cardiovascular disease includes. Basically all laboratory test you mention are tested in the blood/serum or plasma. Please omit using blood ... (e.g. WBC instead of blood WBC). It is heavily affecting the fluidity of the text and adds no information. Propably it makes sense to define it at one point. Ultimately, it is also not consistant through the text/tables.

In page 25 paragraph 2, line 6 we added: “(defined as coronary heart disease and/or cardiac failure)

On pages 26, paragraph 2, line 1 and 27 , paragraph 3, line 1 we added:”(measured in blood or plasma)”. We deleted the word blood or plasma in the rest of the manuscript accordingly.

7. Covid-19 is defined as an infectious disease (does not include asymptomatic SARS-CoV-2 infections). Please change infection to infectious disease.

We changed “infection” for “infectious disease” as suggested.

Introduction:

8. Please cite the RECOVERY trial (contains important and prospective reference information regarding mortality rate and the rate of severe Covid-19 disease). "Can may be used" (page 8) can be formulated a bit more eloquent (e.g. have to potential to be used/applied or can be potentially used). What is mentioned above about decision making applies also here.

In page 8, paragraph 1, line 9 we changed: “...provide a rigorous summary of the evidence available on prognostic factors that can be used in decision-making …” for “...provide a rigorous summary of the evidence available on prognostic factors that may be used in decision-making…”

In page 13, paragraph 1, line 6, we changed: “However, as we identified significant variability in mortality risks reported for similar clinical scenarios we performed a sensitivity analysis using a baseline risk of 26% as reported by a large cohort of non-ICU inpatients treated in 255 sites across 36 countries [6].” for “However, as we identified significant variability in mortality risks reported for similar clinical scenarios (i.e in the RECOVERY trial [20] mortality risk in hospitalized patients assigned to the control arm, with no baseline oxygen requirement was 14%), we performed a sensitivity analysis using a baseline risk of 26% as reported by a large cohort of non-ICU inpatients treated in 255 sites across 36 countries [6].

Methods:

9. The sentence on page 17, chapter 1, line 3 is difficult to understand. "Exposed" is not the correct wording here. The biomarkers/clinical factors/symptoms/etc. are absent or present. It is also not clear if the biomarkers are measured/determined at baseline/SARS-CoV-2 diagnosis or crossed/never crossed the treshold during the complete disease until recovery. Please be more specific in your description. This is essential information!

In page 9, paragraph 3, line 3 we changed: ”We investigated all prognostic factors reported by individual studies and compared patients exposed with patients unexposed to each one of those factors” for “We investigated all candidate prognostic factors reported by individual studies and compared patients exposed (the factor was present) with patients unexposed (the factor was absent) to each one of those factors”

We used the candidate prognostic factors determinations as reported by primary studies authors. How and when those measurements were performed, was a critical part of the risk of bias assessment. Studies in which candidate prognostic factors were not measured at baseline were judged as moderate or high risk of bias.

In page 10 paragraph 2, line 3, we changed: “ We used the Quality in Prognosis Studies (QUIPS) tool for prognostic factor studies [11].” for “ We used the Quality in Prognosis Studies (QUIPS) tool for prognostic factor studies [11] which considers population characteristics, attrition, prognostic factor and outcome measurement and potential residual confounding.”

In S1 Text, page 3, “Risk of bias”, we changed: “We used the Quality in Prognosis Studies tool (QUIPS) for prognostic factor studies.11 To be rated as low risk of bias studies needed to be prospective, have appropriately assessed prognostic factors and outcomes…” for “We used the Quality in Prognosis Studies tool (QUIPS) for prognostic factor studies.11 To be rated as low risk of bias studies needed to be prospective, have appropriately assessed prognostic factors (measured at baseline) and outcomes…” 

10. On page 17, chapter 2, line 5 please change studies results to study results.

We made the suggested modification

11. That you promote an update (AI based) is a huge plus point and might already be necessary!

We acknowledge that an update will provide additional valuable information. However our review includes a significant body of evidence (207 studies including 75607 patients) and we found moderate/high certainty of the evidence for 49 prognostic factors. This means that it is implausible that additional information will modify our conclusions on those factors. We are planning an update which will be focused on the remaining 47 candidate variables for which we found low or very low certainty of the evidence. 

12. The mortality has currently dropped to 5%. Your reference value is no longer up to date!!!

As we mention in the paper we found significant variability in mortality estimates. In order to avoid overestimating the predictive value of the analyzed variables, we decided to use a conservative one which also focused on our population of interest (severe but non-critical patients). We are not aware of a universal significant drop in mortality, for example ISARIC registry reported 26% mortality for non-ICU inpatients until May and 27% mortality for the same population until August (https://isaric.tghn.org/). We think that 9% mortality for severe but non-critical inpatients is reasonable, even if some lower estimates have been reported since our initial analysis. 

13. On page 13, line 2 please change "inseverty" to "in severity".

The term “inseverity” is not included in the manuscript anymore 

Although the cut-offs used are arbitrary, they seem to be weel chosen (e.g. 0.5% in overall mortality is one tenth of the current overall mortality).

Results:

14. Please change "Physical examination factors" to a more appropriate term. Fatique is rather subjective and a symptom. Personally I am also not aware of a test for myalgia. Rewording is required for example to "symptoms, vital signs & abnormal physical examination results".

All along the manuscript we changed: “Physical examination” for “Symptoms, vital signs and physical examination”

15. Laboratory factors: as mentioned above.

Some terms (e.g. immuncompromise) need a better definition (lymphopenia, immunosupressive medication, etc. ???).

In S3 Table, immunosuppression line, we added:”Definition: As defined by the authors including patients on immunosuppressive medications and/or with immunosuppressive medical conditions”

16. Please mention why you didn't look at co-medication! This might be the only factor, which is relevant for decision making!!!

Our systematic review focuses on prognostic/predictive factors for which we used a analytical framework specifically designed to summarize this type of evidence.1 Addressing the effects of interventions (previous or new) requires a different analytical approximation (https://gdt.gradepro.org/app/handbook/handbook.html). These two approximations have substantial differences for example in the process of assessing the certainty of the evidence. A central point is that issues of co-intervention are pertinent to causal associations while for prognostic factors we are merely interested in associations. We then decided not to include interventions in our review. 

17. Overall it is an important review, which summarizes the current biomarker landscape for Covid-19. The cut-off date (End of April 2020) is very early and new discoveries have been made since. NLR and lymphopenia have been even published earlier. They should actually have met the inclusion criteria, but are not included (Lancet, Lancet Resp. Med. Feb 2020). Data have been homgenised and adjusted. The tables, figures and supplementary figures are very informative and have been nicely prepared!

Please see reply to point 10. 

Regarding NLR and lymphopenia, we found moderate certainty that lymphopenia is probably a prognostic factor for mortality and severe disease (page 26, paragraph 2, line 10) and high neutrophil count is probably a prognostic factor for severe disease (page 28, paragraph 1, line 1). Aiming for simplicity as we addressed a significant number of variables, we decided to include candidate prognostic factors only as standalone measurements and not in combination (i.e NLR).

Reviewer #2: This study sets out to report a very important issue: prognostic factors that may be used in decision-making related to the care of patients infected with COVID-19. Although, this topic is not novel, the current research investigated it in a much more detail way than previous studies. In addition, authors also raised the possible questions that must be tackled in near future in order to develop multivariable prognostic models that could eventually facilitate decision-making and improve patients’ outcomes.

Reviewer’s comments:

18. The definitions of severe COVID-19 disease are adopted from multiple studies and therefore are inconsistent. How do the authors eliminate the effect of inconsistent definitions on the association analysis?

We acknowledge that using different definitions of severe COVID-19 disease may have introduced some degree heterogeneity to our results. In response to the reviewers comment we performed subgroup analyses accounting for COVID-19 severe disease definition for 43 of the candidate prognostic factors in which primary analysis had shown significant inconsistency. Observed heterogeneity could not be explained by this analysis for any of the candidate prognostic factors. 

In page 11, paragraph 3, line 6 we added: ”In addition, when we observed inconsistent results for disease severity outcome, we performed subgroup analyses accounting for outcome definition (i.e severity scale vs. IVM vs. ARDS) as a potential source of heterogeneity.”

In page 29, paragraph 4, line 1 we added: “When we found inconsistent results for severe COVID-19 disease outcome (n=43), we performed subgroup analyses accounting on outcome definition. Observed heterogeneity could not be explained by this analysis for any of the candidate prognostic factors explored (see S7 appendix).”

19. When severe COVID-19 disease was not reported as an outcome, the authors considered ICU requirement, invasive mechanical ventilation (IVM) and acute respiratory distress syndrome (ARDS) as surrogate outcomes. Did authors analyze the correlation between primary outcome and surrogate outcomes in this study?

Unfortunately we did not address the correlation between disease severity scales and ICU requirement, IVM or ARDS. However most studies (almost all) used only one of these variables to assess disease severity.

20. The ICU requirement, invasive mechanical ventilation (IVM) and acute respiratory distress syndrome (ARDS) were used as surrogate outcomes. As a results, the outcomes in this study include mortality, severe COVID-19 disease, ICU requirement, invasive mechanical ventilation (IVM) and acute respiratory distress syndrome (ARDS). Multiple outcomes make the results difficult to be interpretated. If those outcomes could be analyzed separately, the results can be interpretated more precisely.

We decided to condense those definitions in one outcome (“Disease severity”) mainly for two reasons: 1) We think that providing separate estimates according to the severity definition used would have resulted in a less clear and possibly more confusing message; 2) All utilized definitions are highly related to each other and reflect severity to a greater or lesser extent, hence a significant effect modification depending on which severity definition was used seemed implausible to us. The latter is supported by the analysis performed in response to comment 17.

21. In the analysis of multivariable models, the variables used for model adjustment include age, one comorbidity (e.g diabetes) and one parameter of disease severity (e.g. respiratory rate) at minimum. How do the authors validate that this setting is statistically appropriate?

In the absence of appropriately validated multivariable prognostic models to rely on,2 we used our clinical judgment and background knowledge to define the thresholds for risk of bias assessment. We acknowledge this as a limitation of our work. In page 31, paragraph 3, we added: “Finally, for risk of bias assessment, we defined a set of requirements for dealing with potential confounders that were not previously validated.”

22. In this study, the authors arbitrarily set thresholds to define important incremental increase in the risk of their outcomes, including mortality or severe COVID-19. How do the authors validate that those thresholds can select proper prognostic information?

We did not validate the thresholds used for result interpretation. We acknowledge that those thresholds would vary depending on multiple factors (e.g. the context and the availability of useful treatments). Although we mention this as a limitation in page 31 we think that being explicit about how the result interpretation process was carried out provides transparency to the process and gives the reader the chance of using alternative interpretation approaches. 

23. The thresholds were set in 0.5% increase in mortality and 1% increase in severe COVID-19 disease. Do the authors analyze the correlation between 0.5% increase in mortality and 1% increase in severe COVID-19 disease?

We did not analyze the correlation between mortality and severe COVID-19 disease in this review. 

24. The authors used the clinical scenario of a patient infected with COVID-19 with severe but not critical disease to assess the prognostic value on mortality. Please clarify the definition of critical disease.

In page 13, paragraph 2, line 3 we added: ”(i.e patients with respiratory failure but not invasive mechanical ventilation and/or hemodynamic support requirement)”

25. When prevalence of prognostic factors was not available, the authors used described baseline risks (9% for mortality and 13 % for severe COVID-19 disease). This setting will inevitably lead to bias in the results.

In page 31, paragraph 3, line 7, we added: ”In addition for some candidate prognostic factors, baseline risk could not be adjusted for prevalence which might have resulted in an overestimation or underestimation of the risk difference estimates.”

26. This study enrolled 207 studies and only 7 were judged as low risk of bias as the remaining presented important limitations in at least one domain or item. Hence, there are multiple inevitable bias in the current study, limiting the reliability of the results.

We agree that the body of evidence has important limitations. Nevertheless exploring how and to what extent those limitations impact the certainty of the observed results is something that requires an extensive analytical process. We have performed that process following state of the art guidance and tools.1 As a result we provide certainty estimates for every candidate prognostic factor and outcome. 

27. There is too much information in Table 1. Is there any way to condense the information or split it into different tables?

We agree. We think that all the information included in table 1. is relevant. We are open to suggestions if the editors consider a better way to present it. For example we could split the table according to the different groups of candidate variables.

28. The authors described that clinicians can use their results to tailor management strategies for patients with COVID-19. It would be of great help to the reader if the authors could formulate a summarized management strategy based on the results of this study.

In page 33, paragraph 2, line 9 we added: ”For example, they could select a set of prognostic factors for which there is high certainty in a significant risk incremental increase (e.g Age, gender, comorbidities, respiratory failure and myocardial injury) and use them to define hospitalization rules for patients consulting to the emergency department.”. We have not discussed specific management strategies, given this is the beyond the scope of our review.

References

1. Foroutan, Farid, Gordon Guyatt, Victoria Zuk, Per Olav Vandvik, Ana Carolina Alba, Reem Mustafa, Robin Vernooij, et al. 2020. “GRADE Guidelines 28: Use of GRADE for the Assessment of Evidence about Prognostic Factors: Rating Certainty in Identification of Groups of Patients with Different Absolute Risks.” Journal of Clinical Epidemiology 121 (May): 62–70. https://doi.org/10.1016/j.jclinepi.2019.12.023.

2. Wynants, Laure, Ben Van Calster, Gary S Collins, Richard D Riley, Georg Heinze, Ewoud Schuit, Marc M J Bonten, et al. 2020. “Prediction Models for Diagnosis and Prognosis of Covid-19: Systematic Review and Critical Appraisal.” BMJ, April, m1328. https://doi.org/10.1136/bmj.m1328.

---

## [Editor Report · Decision Letter 1]

26 Oct 2020

Prognostic factors for severity and mortality in patients infected with COVID-19: A systematic review

PONE-D-20-20042R1

Dear Dr. Izcovich,

We’re pleased to inform you that your manuscript has been judged scientifically suitable for publication and will be formally accepted for publication once it meets all outstanding technical requirements.

Kind regards,

Chiara Lazzeri

Academic Editor

PLOS ONE
---

## [Editor Report · Acceptance letter]

30 Oct 2020

PONE-D-20-20042R1 

Prognostic factors for severity and mortality in patients infected with COVID-19: A systematic review 

Dear Dr. Izcovich:

I'm pleased to inform you that your manuscript has been deemed suitable for publication in PLOS ONE. Congratulations! Your manuscript is now with our production department. 

Kind regards, 

on behalf of

Dr. Chiara Lazzeri 

Academic Editor

PLOS ONE